# Q-GaLore: Quantized GaLore with INT4 Projection and Layer-Adaptive Low-Rank Gradients

Zhenyu Zhang[1], Ajay Jaiswal[1], Lu Yin[2], Shiwei Liu[3], Jiawei Zhao[4], Yuandong Tian[4]*, Zhangyang Wang[1]
[1]University of Texas at Austin, [2] University of Surrey, [3]University of Oxford, [4]Meta AI

Training Large Language Models (LLMs) is memory-intensive due to the large number of parameters and associated optimization states. GaLore [1], a recent method, reduces memory usage by projecting weight gradients into a low-rank subspace without compromising performance. However, GaLore relies on time-consuming Singular Value Decomposition (SVD) operations to identify the subspace, and the frequent subspace updates lead to significant training time overhead. Moreover, GaLore offers minimal improvements in accuracy and efficiency compared to LoRA in more accessible fine-tuning scenarios. To address these limitations, we introduce **Q-GaLore**, a novel approach that substantially reduces memory usage by combining quantization and low-rank projection, surpassing the benefits of GaLore. Our method is based on two key observations: (i) the gradient subspace exhibits diverse properties, with some layers converging early in training while others are subject to frequent changes; (ii) the projection matrices are highly resilient to low-bit quantization. Leveraging these insights, Q-GaLore adaptively updates the gradient subspace based on its convergence statistics, achieving comparable performance while significantly reducing the number of SVD operations. We maintain the projection matrices in INT4 format for aggressive memory conservation and preserve weights in INT8 format, incorporating stochastic rounding to capture accumulated gradient information. This approach enables a high-precision training trajectory using only low-precision weights. We demonstrate that Q-GaLore achieves highly competitive pre-training and fine-tuning performance with exceptional memory efficiency. *At pre-training*, Q-GaLore facilitates training a **LLaMA-7B** model from scratch on a single NVIDIA RTX 4060 Ti with only **16 GB memory**, showcasing its exceptional memory efficiency and practicality. *At fine-tuning*, it reduces memory consumption by **up to 50%** compared to LoRA and GaLore, while consistently outperforming QLoRA (by **up to 5.19** on MMLU) at the same memory cost. Codes are available at `https://github.com/VITA-Group/Q-GaLore`.

## 1. Introduction

Since the 2020s, Large Language Models (LLMs) have demonstrated remarkable performance in various disciplines [2–7]. However, the immense scale of LLMs, often comprising billions of parameters, presents a formidable challenge for most research groups in terms of training and full fine-tuning. For example, Meta's LLaMA models were developed with 2048 A100-80GB GPUs for approximately a period of 5 months [8]. Even without any considerations for product efficiency, fine-tuning a LLaMA 7B model with 16-bit precision necessitates at least 56 GB memory for maintaining the model weight, Adam optimizer states and weight gradient, which is prohibitively expensive.

Numerous research efforts have been dedicated to alleviating the substantial costs associated with training LLMs. These endeavors encompass a range of techniques, including small-scale LLM designing [9, 10], efficient scaling optima [11], training methodologies incorporating sparsity [12–14], sparse model training approaches [15, 16], and low-rank training strategies [1, 17]. Among these, GaLore [1] has emerged as a notable contender, enabling the full-parameter training of LLMs through low-rank gradient updates achieved via Singular Value Decomposition (SVD). Leveraging

---

*Yuandong Tian served as an advisor for this work. All experiments are conducted at the university.

Second Conference on Parsimony and Learning (CPAL 2025).

its low-rank characteristics, GaLore offers a significant reduction—up to 63.3%—in total training memory requirements, facilitating the training of a 7B model with a mere 24GB of memory.

Although GaLore offers substantial memory savings, its 24GB memory requirement still surpasses the available resources in many customer devices. For instance, popular laptop GPUs like the RTX 4060 Ti are equipped with **up to 16GB** of memory. And the price of 24GB RTX 4090 is three times than 16GB RTX 4060 Ti. This limitation raises the question of how we can further reduce the memory footprint of low-rank LLM training to make it accessible to a wider range of hardware configurations. Also, GaLore requires regular updates to the gradient subspace through computationally expensive SVD operations (e.g., every 200 iterations) to approximate the training trajectory of full-rank training. The computational complexity of SVD operations is roughly on the magnitude of $O(mn^2)$, where $m$ and $n$ are the dimensions of the matrix. As a result, it takes $\sim 10$ minutes for the LLaMA-7B model to update the subspace, leading to significant training latency.

To address these challenges, we delved into the training dynamics of the gradient subspace of GaLore and discovered two intriguing phenomena: (i) The gradient subspace of GaLore demonstrates different behaviors across different layers, in which some layers demonstrates "early bird" properties and converge within the initial training stage while some layers have a stable subspace within a specific window during training and some other layers consistently keeps changing. (ii) The projection matrices of GaLore exhibit excellent quantization-friendliness property, which can be seamlessly quantized to 4-bits without sacrificing training quality.

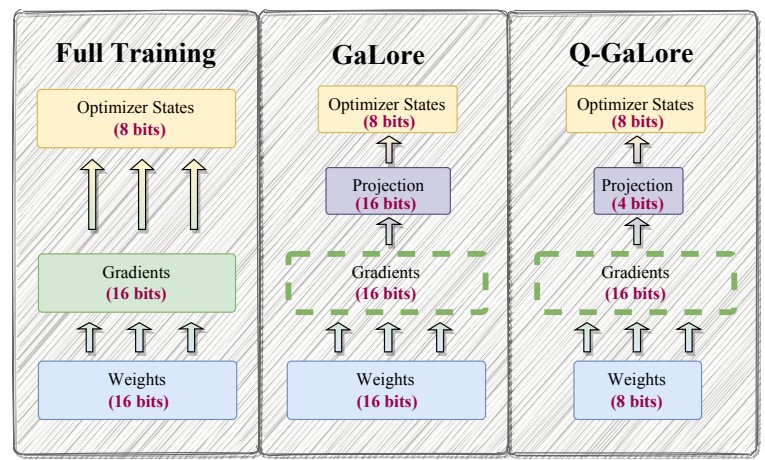

Figure 1: Comparison of data types and training flows of different methods. We by default use 8-bits Adam [18] as the inner optimizer. The gradient in GaLore and Q-GaLore is not persistent during training.

Inspired by these observations, we propose Q-GaLore, a novel approach that enables the training of large language models with low-precision weights and low-rank gradients. Q-GaLore introduces two modules to reduce memory overhead and training latency:

(i) **Low precision training with low-rank gradients**: We manage to quantize the entire model (not only the optimizer state as in GaLore [1]) to 8-bits and the projection matrix to **4-bits**, as shown in Figure 1. By utilizing low-precision weights and projection matrices, our approach achieves a reduction of approximately 28.57% in memory requirements for gradient low-rank training where the weight represent the primary component of memory usage post low-rank projection. Additionally, to maintain training stability and approximate the trajectory of high-precision training, we implement Stochastic Rounding (SR) [19] that provides an unbiased estimation of the gradient trajectory and mitigates gradient information loss, thus enhance the training stability and overall performance.

(ii) **Lazy layer-wise subspace exploration**: We monitor the convergence levels of the gradient subspace in different layers and adaptively decrease the frequency of SVD operations for the layers whose low-rank subspace does not change significantly over time. This approach reduces the training time associated with SVD, saving over 32 hours for training a 7B model.

We demonstrate the efficacy of Q-GaLore in both pre-training and fine-tuning scenarios. For pre-training, Q-GaLore's efficiency allows us to reduce the memory requirements of full-rank training and GaLore by 61% and 30%, respectively, across various model sizes from 60M to 7B. Notably,

Q-GaLore demonstrates the feasibility of training LLaMA-7B on a single NVIDIA RTX 4060 Ti with only 16GB of memory while significantly reducing memory costs when using data parallism for large batch training. For fine-tuning, Q-GaLore matches the performance of SOTA low-rank approaches including LoRA [20], QLoRA [21] and GaLore [1]. It reduces memory consumption by up to 50% than LoRA/GaLore, while consistently outperforming QLoRA [21] at the same memory cost.

## 2. Related Work

### 2.1. Low-Rank Adaptation and Training

Optimizing Large Language Models (LLMs) requires a substantial memory footprint to accommodate weights, activations, gradients, and optimization states. Low-Rank Adaptation (LoRA) [20] is a notable technique that introduces low-rank weight adapters for each layer, reducing the memory footprint by only optimizing the adapters, which can later be merged back into the original model. Subsequent enhancements to LoRA, such as quantization [21], multi-task learning support [22], and various architectural improvements [23–28, 28–30], have all focused on fine-tuning scenarios. Despite the efficiency of low-rank adaptation, its suboptimal performance compared to full parameter optimization [31] has motivated the development of other memory-efficient optimization methods. For instance, [32, 33] reduce memory overhead through fused backward operations, eliminating the need to store all weight gradients. Sparse optimization techniques, such as BAdam [34] and LISA [35], partition parameters into blocks or sample layers based on importance to minimize memory costs while maintaining performance comparable to full parameter fine-tuning.

Early efforts to adapt LoRA for pre-training, such as ReLoRA [36], still require full-rank learning in the initial stages, resulting in high memory overhead. Recently, GaLore [1] leverages the low-rank properties of gradients [28] to enable full-parameter learning while significantly reducing memory usage during optimization. This approach allows GaLore to achieve better performance than common low-rank adaptation methods such as LoRA, while still being memory-efficient.

### 2.2. Low Precision Training

Low-precision training aims to improve training efficiency by storing data in low-precision formats and leveraging low-precision General Matrix Multiplication (GEMM) operations. This is distinct from post-training quantization, which primarily enhances the inference efficiency of pre-trained models. A significant challenge in low-precision training is potential instability during the training process. SWALP [37] addresses this issue using stochastic weight averaging [38], but it requires maintaining averaged weights, leading to high memory overhead in large foundational models. Other methods handle instability by scaling gradients [39] or second-order optimizer statistics [40].

While various low-precision training methods have been explored for smaller-scale convolutional networks [41–46], they are generally not applicable to training large-scale transformers, as large tensors are less suitable for quantization [47]. Some approaches to low-precision training at a larger scale still require maintaining high-precision latent weights during training, significantly increasing memory consumption for large language models [48, 49]. This study aims to improve the end-to-end memory efficiency of training large-scale foundational model at scale.

## 3. Methodology

We first introduce the data type and quantization basics in Section 3.1. Section 3.2 demonstrates the adaptive convergence properties of the gradient subspace, which facilitates efficient training. In Section 3.3, we demonstrate the high tolerance of the projection matrix to quantization. Section 3.4 then discusses stochastic rounding for approximating high-precision training trajectories. The overall pipeline of Q-GaLore is depicted in Figure 4.

### 3.1. Preliminaries on Quantization

Generally, quantization methods are categorized into Post-Training Quantization (PTQ), where quantization is applied to pretrained models without further training; and Quantization-Aware Training (QAT), which incorporates quantization throughout the training process. QAT aims to either generate more quantizable models for faster inference or expedite the training process through low-precision operations. To preserve performance, these methods retain high-precision parameters throughout the training process and apply quantization to transfer the parameters into low-precision data formats during each forward and backward pass. Maintaining high precision parameters occupis massive memory and results in even larger memory requirements than vanilla high precision training. In this work, we focus on improving the memory efficiency of training large language models and do not maintain the high-precision parameters.

In Q-GaLore, the model weights are retrained in INT8 while activations and gradients are computed in BFloat16. Although FP8 [50] offers greater expressiveness than INT8, it is supported on limited devices, *e.g.*, the NVIDIA Hopper series GPUs, which are costly and not widely available. Thus, we employ the more general INT8 formats. The pseudocode is presented in the appendix A. To convert data format, we utilize block-wise uniform quantization [51]:

$$W_q = \text{Quant}_n(W, s, z) = \text{clamp}(\lfloor \frac{W}{s} \rceil + z, -2^{n-1}, 2^{n-1} - 1)$$

where $W$ and $W_q$ represents the original and quantized tensors, respectively. $s$ is the scaling factor and $z$ is the zero point. Both $s$ and $z$ are calculated within each block of the tensors. $n$ is the quantization bits. We default to use block size of $256$ in all implementations.

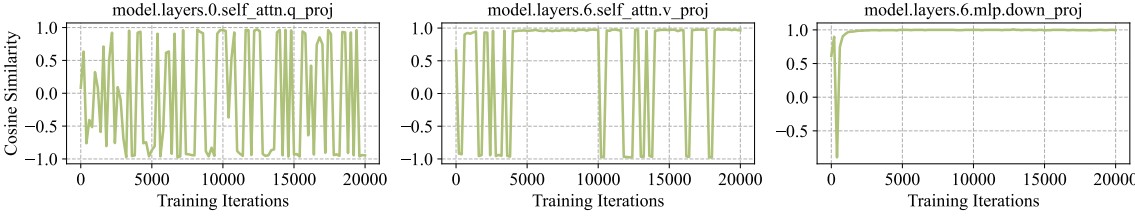

Figure 2: Cosine similarity between the adjacent projection matrices captured every 250 training iterations.

### 3.2. Layerwise Convergence Behaviors of Gradient Subspace

GaLore relies on a fixed interval to recompute the gradient space and projection matrices blindly, assuming that the training dynamics of all the layers in LLMs remain the same. One direct implication remains the frequent computation of computationally expensive SVD. Inspired by previous work about training dynamics of gradient subspace [52], we're curious: *How does the gradient subspace dynamics varies during the pre-training of LLMs?* We investigated the cosine similarity across the projection matrices obtained at regular interval during the pre-training of LLaMa-130M as shown in Figure 2. Our observations are as follows: (i) certain layers exhibit an "early bird" phenomenon, whereby their gradient subspace saturates early during pre-training and remains stable throughout (Top Right, with cosine similarity close to 1); (ii) in some layers, the gradient subspace saturates within a specific window during pre-training (Top Middle); (iii) for others, the gradient subspace consistently keeps changing towards the end of training (Top Left).

This observation provides a unique opportunity to monitor the gradient subspace behavior during pre-training and dynamically update the frequency of SVD for each layer if we observe saturation. More specifically, starting with an SVD interval of $t$ for a layer $l$, we monitor the cosine similarity of projection matrices in the previous $k$ intervals. If the cosine similarity across the k intervals remains greater than a threshold (*e.g.*, $\geq 40\%$), we update the interval from ($t \to 2 \times t$) to reduce the compute. This **adaptive lazy update** can closely mimic the performance of the original GaLore with over

60% reduction in computationally expensive SVD calls. Further ablation studies about the trade-off between SVD calls and performance are presented in Section 4.4.

## 3.3. High Quantization Tolerance of Projection Matrix

The adaptive convergence properties suggest that the projection matrix has a degree of redundancy, indicating that high accuracy is not essential. This observation inspired us to further investigate the functionality of the projection matrix under quantization conditions. We implemented block-wise quantization for the projection matrices, maintaining a uniform block size of 256 across all layers. During these experiments, we ensured that the update steps for the projection matrices remained constant, allowing us to focus exclusively on their quantization characteristics. Figure 3 illustrates the results for the LLaMA-130M models, demonstrating that the projection matrices are highly resilient to

Figure 3: Pre-training performance on the LLaMA-130M models. The projection matrices are quantized with different bits.

quantization, with minimal impact on pre-training quality even when reduced to 4 bits. Based on these findings, we applied quantization to the projection matrices, restricting them to 4 bits. This approach further reduces the memory cost of the optimizer states in low-rank training by 25%.

## 3.4. Approximating High-Precision Training Trajectories by Stochastic Rounding

When using low-rank training methods such as GaLore, the allocation of memory to maintain model parameters constitutes the majority of the memory overhead. Consequently, we opt to maintain the weights in low precision to enhance memory efficiency during training. The primary challenge of training with low-precision parameters is the significant reduction of gradient information. During each optimization step, the full precision gradient must be quantized to a low precision weight update. However, if the gradient magnitude is not large enough, it will be mitigated via the round-to-nearest scheme. Conventional Quantization-Aware Training (QAT) retains full precision parameters to accumulate small gradient contributions, albeit at the cost of increased memory overhead. To address this issue, we employ Stochastic Rounding (SR) [19, 53, 54], that is formulated as the following:

$$W_q = \mathcal{F}_{SR}(W) = \begin{cases} \lfloor W \rfloor & \text{with probability } p = \lceil W \rceil - W \\ \lceil W \rceil & \text{with probability } p = W - \lfloor W \rfloor \end{cases}$$

Under this formulation, the expected value of $W_q$ is $E[W_q] = \lfloor W \rfloor (\lceil W \rceil - W) + \lceil W \rceil (W - \lfloor W \rfloor) = W$, allowing the low-precision parameters to implicitly accumulate small gradient information. This method achieves comparable performance without the substantial memory requirements associated with maintaining high-precision parameters.

## 3.5. The Q-GaLore Algorithm

The pipeline of Q-GaLore is illustrated in Figure 4. The left section of the figure depicts the computation flows, where only the gradients are maintained in high precision to preserve essential training dynamics information. We employ an 8-bit version of the Adam optimizer [18] as the internal optimizer. During each training iteration, the full-rank gradient is projected into a low-rank format and then incorporated into the optimizer states. To project the gradient into the subspace, we obtain the projection matrix using Singular Value Decomposition (SVD), as described in [1]. The update frequency of the projection matrix is managed through our adaptive update strategy, and the matrix is quantized to 4-bits formats to reduce memory overhead.

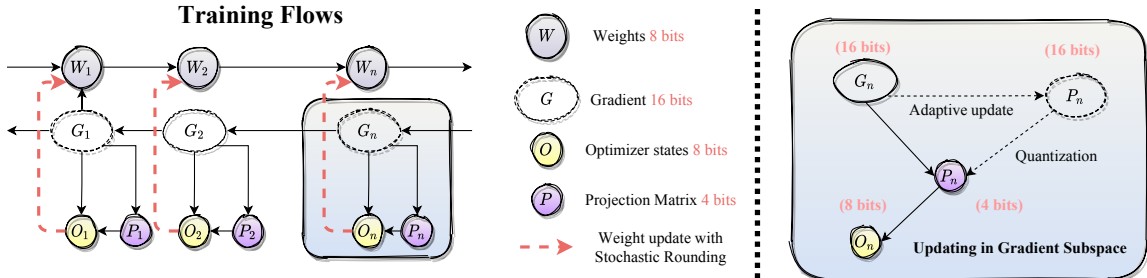

Figure 4: Illustration of the training flows for Q-GaLore, where the dotted icon denotes intermediate tensors that do not consistently occupy memory.

Furthermore, after updating the optimizer states, we project the low-rank optimizer states back to full rank and update the parameters. As the weights are consistently maintained at low precision, an additional quantization step is necessary to update the weights. Here, we utilize SR to capture the minor-gradient nuances and provide an unbiased estimation of the high-precision weights. And we employ a fused backward operation as described in [1, 32, 33] when gradient accumulation is disabled. Upon calculating the gradients for a single layer, we promptly update the corresponding optimizer state and weights, subsequently releasing the memory allocated to the gradients. If gradient accumulation is required, we then accumulate the gradient in the low-rank format, resulting around one quarter memory consumption of full gradient accumulation.

# 4. Experiments

## 4.1. Implementation Details

**Network Architecture.** For the pretraining task, we adopt the LLaMA-based architecture with sizes ranging from 60 million to 1 billion, following the setups from [1, 36]. During downstream experiments, we select various pre-trained models to evaluate the general effectiveness of Q-GaLore, including RoBERTa [55] base, LLaMA-3-8B [56], Gemma-7B [57], and Mistral-7B [58].

**Pre-Training.** We pre-train the LLaMA models on C4 dataset [59]. The C4 dataset is a massive collection of Common Crawl's web crawl corpus, meticulously filtered and cleaned to ensure high-quality language modeling and training. It is widely used for pre-training large language models due to its diverse and extensive textual content. We train the models on this sufficiently large dataset without data repetition.

**Fine-Tuning.** The downstream tasks cover two categories: (i) GLUE benchmarks [60], a series of widely used tasks for evaluating the downstream performance of natural language understanding; (ii) MMLU [61] that evaluates the natural language understanding ability of LLMs, covering various domains, including STEM, social sciences, humanities and others.

**Baselines.** We consider five baseline methods for comparison: (i) `Full`: Models are trained with the original Adam [62] optimizer. Both weights, gradients, and optimization states are maintained with full rank and full precision (BF16 format). (ii) `Low-Rank`: The original weights are factorized into low-rank components: $W = UV$, and $U$ and $V$ are optimized via Adam [63]. (iii) `LoRA`: LoRA [20] introduces low-rank adaptors for training the models, $W = W_0 + UV$, where $W_0$ is the pretrained weights, which are frozen during training. We use the initialized weight as $W_0$ during pretraining and only optimize $U$ and $V$. And we default to 32 for LoRA alpha and 0.05 for LoRA dropout. (iv) `ReLoRA`: ReLoRA [36] enhances the original LoRA methods for better pre-training. ReLoRA is a stage-wise LoRA that periodically merges $UV$ into the original $W$ and initializes a new $UV$ for continued training. (v) `QLoRA` [21]: we use the same hyperparameters: 32 for QLoRA alpha and 0.05 for QLoRA dropout. We keep the base models in 8bits for fair comparison. (vi) `GaLore` [1]: We project the

gradient into low-rank format and update the optimizer states. When updating the weight, we project back the low-rank weight update to full-rank. We follow the original hyperparameters, setting the subspace frequency in GaLore to 200 and the scale factor $\alpha = 0.25$. The low-rank dimension is chosen as a quarter of the original dimension. Note that all baseline methods, except QLoRA, are maintained in 16-bit precision, while the base models in QLoRA are kept in 8-bit precision for a fair comparison.

## 4.2. End-to-End Results

### 4.2.1. Memory-Efficient Pre-training with Q-GaLore

We pre-trained the LLaMA-based models from scratch on the C4 dataset using various memory-efficient methods. The experiments encompassed different model sizes ranging from 60 million to 1 billion parameters, with results reported in Table 1. In each experiment, we report the perplexity values obtained on the validation set. As the primary memory savings are derived from compressing the weight and optimizer states, we provide estimates of the memory overhead associated with storing these components. Detailed discussions on end-to-end memory measurements and throughput comparisons are provided in Section 4.3. For fair comparison, we used the same low-rank dimensions for all the memory-efficient approaches, specifically {128, 256, 256, and 512} for {60M, 130M, 350M, and 1B} models, respectively. And we use 16-bits Adam as the inner optimizer inside GaLore while Q-GaLore implements 8-bit Adam optimizer.

Incorporating adaptive subspace updating, projection and weight quantization, and stochastic rounding, our Q-GaLore method maintains comparable pre-training performance (with less than a 0.84 perplexity increase, compared with the original GaLore approach) while significantly reducing memory overhead. For example, in the experiment of 1 billion model size, training with INT8 weights halved the original memory cost for weights and achieved a 29.68% memory saving against the original GaLore method and a 60.51% memory saving compared to the Full baseline. Compared to GaLore, the additional memory savings primarily come from two sources: (i) INT8 weights require only half the memory overhead of BF16 weights, and (ii) INT4 projection matrices reduce approximately 25% of the memory overhead for optimization states.

Table 1: Comparison results of various memory-efficient algorithms on pre-training tasks. Experiments are conducted on C4 dataset with LLaMA models. For each experiment, we report both the perplexity and estimated memory. The estimated memory only count for the weights and optimizer states which cost the majority memory overhead. We follow the same settings and collect the results of all baseline methods from [1], where the training tokens are {1.1B, 2.2B, 6.4B, 13.1B} for {60M, 130M, 350M, 1B} models, respectively.

| Methods | 60M | | 130M | | 350M | | 1B | |
|---|---|---|---|---|---|---|---|---|
| | Perplexity | Memory | Perplexity | Memory | Perplexity | Memory | Perplexity | Memory |
| Full | 34.06 | 0.36G | 25.08 | 0.76G | 18.80 | 2.06G | 15.56 | 7.80G |
| Low-Rank | 78.18 | 0.26G | 45.51 | 0.54G | 37.41 | 1.08G | 142.53 | 3.57G |
| LoRA | 34.99 | 0.36G | 33.92 | 0.80G | 25.58 | 1.76G | 19.21 | 6.17G |
| ReLoRA | 37.04 | 0.36G | 29.37 | 0.80G | 29.08 | 1.76G | 18.33 | 6.17G |
| GaLore | 34.88 | 0.24G | 25.36 | 0.52G | 18.95 | 1.22G | 15.64 | 4.38G |
| Q-GaLore | 34.88 | 0.18G | 25.53 | 0.39G | 19.79 | 0.88G | 16.25 | 3.08G |

### 4.2.2. Memory-Efficient Fine-Tuning with Q-GaLore

Pre-training LLMs is a resource-intensive task that is typically only feasible for large companies or computing centers. In most practical scenarios, memory-efficient fine-tuning of LLMs on specific downstream tasks is more common. To evaluate the effectiveness of Q-GaLore, we selected a diverse set of downstream tasks, including eight tasks from the GLUE benchmark and four subtasks from MMLU, which assess the ability of LLMs to understand natural language. We compared the performance of Q-GaLore with the baseline Full method and three state-of-the-art low-rank optimization approaches: LoRA, GaLore and QLoRA. It is important to note that while GaLore utilizes

a 16-bit Adam optimizer, Q-GaLore employs an 8-bit Adam optimizer, further reducing memory requirements without compromising performance.

Table 2: Comparison results of various memory-efficient fine-tuning algorithms on MMLU tasks. Note that the reported memory stands for the estimated memory overhead for weights and optimizer states. End-to-end memory measurements are discussed at Section 4.3.

| Model | Methods | Memory | STEM | Social Sciences | Humanities | Other | Average |
|---|---|---|---|---|---|---|---|
| LLaMA-3-8B | Full | 48 GB | 54.27 | 75.66 | 59.08 | 72.80 | 64.85 |
| | LoRA | 16 GB | 53.00 | 74.85 | 58.97 | 72.34 | 64.25 |
| | GaLore | 16 GB | 54.40 | 75.56 | 58.35 | 71.19 | 64.24 |
| | QLoRA | 8 GB | 53.63 | 73.44 | 58.59 | 71.62 | 63.79 |
| | Q-GaLore | 8 GB | 53.27 | 75.37 | 58.57 | 71.96 | 64.20 |
| Gemma-7B | Full | 51 GB | 30.03 | 37.16 | 34.08 | 35.47 | 34.21 |
| | LoRA | 17 GB | 26.23 | 34.94 | 30.88 | 36.96 | 32.18 |
| | GaLore | 17 GB | 27.33 | 36.74 | 30.82 | 37.90 | 33.20 |
| | QLoRA | 9 GB | 24.83 | 27.54 | 28.09 | 33.40 | 28.49 |
| | Q-GaLore | 9 GB | 27.73 | 36.80 | 32.54 | 37.89 | 33.68 |
| Mistral-7B | Full | 43 GB | 52.40 | 72.95 | 55.16 | 69.05 | 61.67 |
| | LoRA | 14 GB | 52.13 | 72.46 | 55.05 | 68.77 | 61.41 |
| | GaLore | 14 GB | 51.50 | 73.02 | 55.03 | 69.49 | 61.55 |
| | QLoRA | 7 GB | 50.00 | 71.29 | 55.84 | 67.66 | 60.70 |
| | Q-GaLore | 7 GB | 52.23 | 72.82 | 55.01 | 69.30 | 61.62 |

Table 3: Comparison results of various memory-efficient fine-tuning algorithms on GLUE tasks, with the pretrained RoBERTa model (Note that the number of training epochs used in the original LoRA [20] work is higher than that in GaLore [1]. To ensure consistency, we maintain the same training settings as in [1] and use its baseline results for comparison). We report the Matthew's correlation for the CoLA task, Pearson correlation for STS-B, average (matched and mismatched) accuracy for MNLI, F1 score for MRPC, and accuracy for all other tasks. The reported memory stands for the estimated memory overhead for weights and optimizer states. End-to-end memory cost are discussed at Section 4.3.

| Methods | CoLA | STS-B | MRPC | RTE | SST2 | MNLI | QNLI | QQP | Average | Memory |
|---|---|---|---|---|---|---|---|---|---|---|
| Full | 62.24 | 90.92 | 91.30 | 79.42 | 94.57 | 87.18 | 92.33 | 92.28 | 86.28 | 747 MB |
| LoRA | 60.06 | 90.82 | 92.01 | 79.78 | 94.38 | 87.17 | 92.20 | 91.11 | 85.94 | 264 MB |
| GaLore | 61.83 | 90.80 | 91.90 | 79.06 | 93.46 | 86.94 | 92.25 | 91.22 | 85.93 | 257 MB |
| QLoRA | 60.16 | 89.93 | 91.87 | 71.84 | 93.92 | 86.57 | 92.29 | 91.17 | 84.72 | 183 MB |
| Q-GaLore | 61.60 | 90.23 | 91.96 | 79.06 | 94.38 | 86.73 | 92.44 | 90.91 | 85.91 | 176 MB |

Tables 2 and 3 lead to consistent observations: (i) Q-GaLore achieves performance comparable to the full fine-tuning baseline across different models (LLaMA-3-8B, Gemma-7B, Mistral-7B, and RoBERTa-base), with a minimal performance gap of less than 0.65 compared to `Full`; (ii) Q-GaLore demonstrates comparable or even superior performance compared to LoRA, with a improvement of 1.02 performance gain on the MMLU benchmark of Gemma-7B while also requiring less memory; (iii) Compared with QLoRA, Q-GaLore demonstrates **consistent (up to 5.19) gains** of performance across architectures and tasks, at the same memory costs.

## 4.3. End-to-End Memory Measurement

We present an end-to-end memory measurement for training a LLaMA-7B model in Figure 5. Starting from the baseline full parameter training with BF16 Adam optimizer, 8-bits Adam optimizer halves the memory overhead of the optimizer states by quantizing them to a lower precision format. Then, 8-bits GaLore further compresses the memory cost by converting the optimizer states into a low-rank format. Moreover, 8-bits GaLore employs a fused backward operation that sequentially releases the gradient

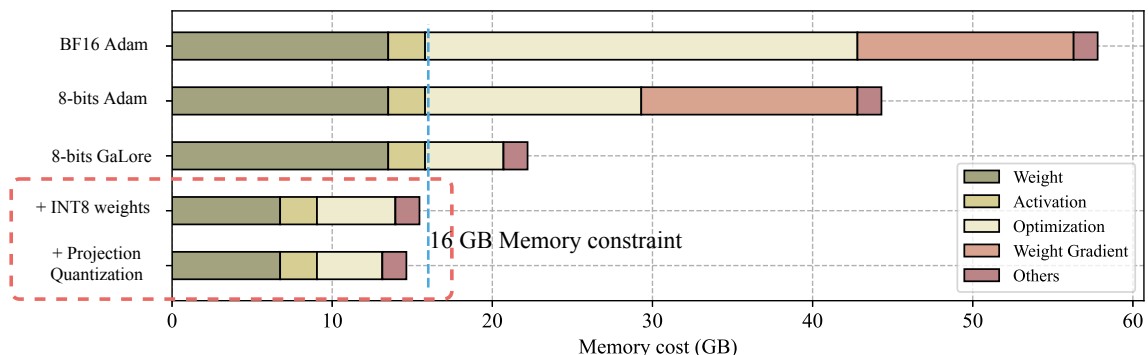

Figure 5: Results of the memory allocation of training a LLaMA-7B model with a single batch size of 256.

memory, rendering the gradient memory cost negligible. Building on this, Q-GaLore incorporates INT8 weights, which halve the memory requirement for weights. Projection quantization then further reduces the memory allocated to optimizer states. Notably, only Q-GaLore can train a LLaMA-7B model within the 16 GB memory constraint, demonstrating the potential for optimizing models on edge devices. Additionally, due to the varying data formats of gradients and weights, the requisite quantization and dequantization operations incur a throughput overhead of $14.64\%$, as compared to the original GaLore. We will improve the implementation for further work. Furthermore, Q-GaLore can enable large batch training when combined with FSDP, significantly reducing the memory consumption of weights and optimizer states on each GPU. This allows for training with fewer GPUs, thereby reducing communication overhead.

## 4.4. Further Investigation and Ablation Study

In this section, we focus on the ablation studies of Q-GaLore, centering on two key questions: *Q1*: How does Stochastic Rounding (SR) benefit the training process? *Q2*: What is the trade-off between training performance and SVD counts in Q-GaLore?

*A1*: **Enhanced low-precision training with stochastic rounding.** Stochastic rounding provides an unbiased estimation of accumulated gradient information, which is crucial for low-precision training. We conducted controlled experiments to pre-train LLMs with and without stochastic rounding. To ensure a fair comparison, we maintained consistency in other hyperparameters across the experiments: weights were stored in the INT8 data format, projection matrices were subjected to 4-bit quantization, and the adaptive convergence ratio for the gradient subspace was set at 0.4.

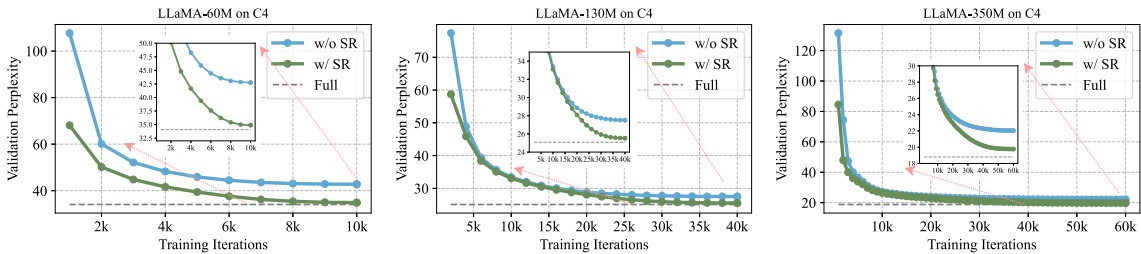

Figure 6: Ablation study of pre-training with Q-GaLore w/ or w/o Stochastic Rounding (SR). `Full` curve stands for the perplexity of the final checkpoint that optimized by original Adam optimizer. Each subfigure includes a smaller inset that represents the zoomed-in results.

Figure 6 illustrates the perplexity on the validation set throughout the training process. At each training step, gradient information is quantized back to the low-precision format (INT8), resulting in considerable information loss and suboptimal performance. The perplexity increased by 7.86, 1.98,

and 2.27 for models with sizes of 60, 160, and 350 million parameters, respectively. Additionally, we implemented an initial warm-up stage for pre-training for training stability, where the weight updates are generally smaller. During this stage, significant loss of gradient information occurs due to the vanilla round-to-nearest scheme, resulting in a perplexity gap ranging from 18.67 to 47.02, compared with models using stochastic rounding. Meanwhile, Q-GaLore can effectively capture the gradient information, achieving performance comparable to the `Full` baseline, with a perplexity gap of less than 1.

*A2*: **Over 60% SVD operations costs can be saved for free.** We explore the trade-off between the number of SVD operations used for updating the gradient subspace and pre-training performance on the LLaMA-130M model. In this study, we perform a grid search for the cosine similarity threshold within the range $[0, 1]$ and report the corresponding SVD counts along with the perplexity. Figure 7 demonstrates that there is an efficient reduction in SVD counts; with only 36.20% of SVD operations, Q-GaLore (where the cosine similarity threshold equals 0.4) can achieve comparable performance to the GaLore baseline, resulting in significant time savings. Specifically, to update the gradient subspace of a LLaMA-7B model, the SVD

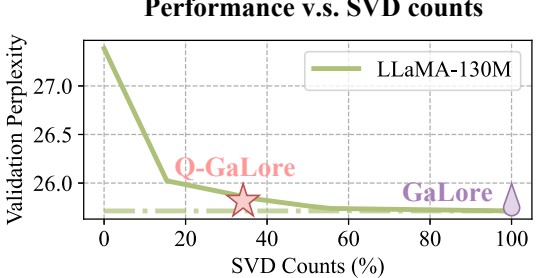

Figure 7: Trade-off between performance and SVD counts for updating gradient subspace. Results are normalized by SVD counts of original GaLore.

operation requires approximately 10 minutes when measured on a single NVIDIA RTX A6000 GPU; and this gradient subspace is updated 300 times across 150,000 training iterations. By achieving more than 60% savings in SVD operations, our method significantly reduces the time cost by over 32 hours. Moreover, Q-GaLore can further enhance the training throughput by allowing larger batch-size training. For example, the reduced memory overhead enables us to train a LLaMA-3B model with a batch size of 32 on two A6000 GPUs, compared to a batch size of 16 supported by GaLore. This results in a throughput improvement from 14.68 to 15.23 samples/second.

## 5. Conclusion

To overcome these challenges and further enhance memory-efficient training, we propose Q-GaLore, a method that reduces memory usage through quantization and low-rank projection. Our approach is motivated by two key observations during gradient low-rank training: (1) the gradient subspace exhibits diverse properties, with some layers converging at the very early training stages while others are subject to frequent changes; (2) the projection matrices demonstrate high quantization-friendliness and function effectively under 4-bit quantization. Building on these, Q-GaLore enables low-precision training (INT8 for the entire model and INT4 for the projection matrix) with low-rank gradients and significantly fewer SVD operations. Our experiment results demonstrate that Q-GaLore achieves competitive performance on both pre-training and fine-tuning tasks.

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

# A. More Implementation Details

The pseudo-code of the forward and backward process in PyTorch style are illustrated in the following:

```python
class INT8Linear(torch.autograd.Function):
    @staticmethod
    def forward(ctx, x, INT8_W):
        ctx.save_for_backward(x, INT8_W)
        W = (INT8_W.to(x.dtype) - INT8_W.zeros) * INT8_W.scales
        return x @ W.t() + bias

    @staticmethod
    def backward(ctx, grad_output):
        x, INT8_W = ctx.saved_tensors
        W = (INT8_W.to(x.dtype) - INT8_W.zeros) * INT8_W.scales
        grad_input = grad_output @ W
        grad_W = grad_output.t() @ x
        return grad_input, grad_W
```

# B. More Experiment Results

Stochastic rounding is an effective strategy to mitigate ineffective weight updates caused by quantization. However, the low-rank gradient projection introduces additional noise into the gradient, potentially leading to greater bias in the rounded gradient compared to full-precision training. To investigate this, we conducted simulation experiments where the full-rank gradient is retained throughout the training process, serving as a calibration for the rounding direction, while the actual weight updates are performed using the low-rank gradient. Experiments were conducted on the LLaMA-130M model with a pre-training task on the C4 dataset, achieving a perplexity of 25.28 on the validation set, with no significant improvement over the original Q-GaLore method, which achieved a perplexity of 25.53. These results suggest that low-rank gradient projection does not diminish the effectiveness of stochastic rounding.

# C. Experiment Hyperparameters

**Details of pre-training on C4** We follow the same setups in GaLore and training the LLaMA with a total batch-size of 512. And the whole training steps are {10000, 20000, 60000, 100000} for {60M, 130M, 350M, 1B} models, respectively. For each experiment, we use a warm-up learning rate strategy in the initial one-tenth training phase and cosine annealing decay in the following. The default base update interval is set to 200 iterations, using the lazy subspace update approach with a cosine similarity threshold of 0.4. The rank of gradient is set as {128, 256, 256, 1024} for {60M, 130M, 350M, 1B} models, respectively.

**Details of fine-tuning on GLUE** We fine-tune the pre-trained RoBERTa-based model for 30 epochs on each task from the GLUE benchmark. The learning rate is set to $1 \times 10^{-5}$ for all tasks, except for MRPC and CoLA, where a learning rate of $3 \times 10^{-5}$ is used. The batch size is set to 32 for CoLA and 16 for all other tasks. And the rank of gradient is fixed at 8.

**Details of fine-tuning on MMLU** For each experiment, we fine-tune the model for 3 epochs with a batch size of 8. And the learning rate is set to $\{1 \times 10^{-5}, 5 \times 10^{-5}, 3 \times 10^{-5}\}$ for {Mistral-7B, LLaMA-3-8B, Gemma-7B}, respectively. We use the cosine annealing scheduler for learning rate decay where the initial one-tenth training steps is used as warm-up. The rank of gradient is kept as 8.

# D. Gradient Subspace of Different Layers

We evaluate the gradient subspace across different layers in Figure 8. We observe that, generally, the q and k projections exhibit more diverse gradient subspaces. This is because q and k are responsible

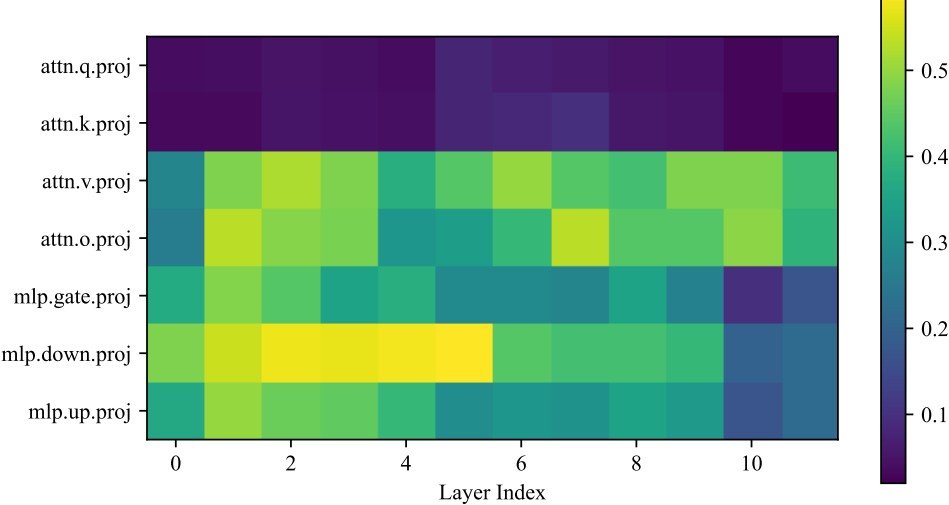

Figure 8: The average cosine similarity of gradient subspace projection on LLaMA-130M, the cosine similarity is calculated across two adjacent projection matrix, and being averaged across the training process.

for generating attention patterns, which heavily depend on different tokens, thereby demonstrating significant diversity. In contrast, the down projection shows the most consistent subspace. Additionally, middle layers tend to have more consistent gradient subspaces compared to the initial and final layers. This behavior might related to the oversmoothing issue in Transformers, where middle layers are not well-optimized are casuing the token representations become oversmoothing.

