# OpenReview forum: "Q-GaLore: Quantized GaLore with INT4 Projection and Layer-Adaptive Low-Rank Gradients"
_CPAL.cc/2025/Proceedings_Track — CPAL 2025 (Proceedings Track) Poster_

### Official Review · Reviewer_nCgZ · 2025-01-09
**Review of Submission60**

**Rating:** 6
**Confidence:** 5

**Review:**

Summary
This paper improves on the recently proposed GaLore method via reduced memory costs and less overhead via quantization and adaptive schedule for updating projections via SVD respectively. Experiments on pre-training and fine-tuning are conducted to show benefits over existing pre-training and fine-tuning methods.

Pros
Paper is well presented with useful figures.
The per-layer adaptive SVD update and quantization are justified via practical experiments.
I find the reduction in SVD operations particularly practical and a major improvement over GaLore.

Cons
Applying quantization in the context of parameter efficient training/fine-tuning is not completely novel (e.g. QLoRA).
Improvements over QLoRA in the fine-tuning setting are relatively limited besides for the Gemma model, is there a reason QLoRA deteriorates in performance to such an extent for this model?

As an aside, I believe the authors should cite the following paper, which similarly shows that the gradient subspace across many layers do not change much in direction from initialization: Yaras et al. Compressible Dynamics in Deep Overparameterized Low-Rank Learning & Adaptation. ICML 2024.

---

### Official Review · Reviewer_XX9m · 2025-01-10
**Improved version of GaLore**

**Rating:** 4
**Confidence:** 4

**Review:**

This paper proposes strategies to improve GaLore's memory efficiency by combining quantization with a lazy SVD update operation. The approach is motivated by two observations in LLM training: (1) projection matrices are robust to quantization, and (2) gradients change at different rates across layers. Extensive experiments on both pre-training and fine-tuning validate the method, demonstrating comparable performance to previous approaches with reduced memory usage.

**Pros:**
* The paper is well-structured, with a solid presentation and thorough discussion of related work.
* The motivations behind the proposed changes are clear, supported by trends visualized in Figures 2 and 8, which highlight implicit properties of LLMs.
* Comprehensive experiments cover various models and both pre-training and fine-tuning scenarios.

**Cons:**
* The choice of hyperparameters (e.g., LoRA alpha set to 32 and dropout to 0.05) deviates from the defaults in the original LoRA paper. The rationale for these changes is unclear. For instance, in Table 3, while the proposed method achieves comparable results to LoRA, the reported LoRA performance is significantly lower than in the original paper (which is reproducible based on my own trial). A more careful and justified comparison is necessary to ensure fairness and make the results demonstrated in the paper more convincing. Additionally, the average values in Table 3 for LoRA/GaLore are incorrect and should be 85.93/85.94.
* Line 51, the authors state that "it takes ~10 minutes for the LLaMA-7B model to update the subspace..." for GaLore, I'm wondering what would be the time usage for Q-Galore in this scenario.

---

### Official Review · Reviewer_rGMP · 2025-01-10

**Rating:** 8
**Confidence:** 5

**Review:**

## Summary
This work proposed a quantized variant of the recent GaLore algorithm. Specifically, Q-GaLore quantizes the network weights to int8 and the gradient projection matrices to int4. Motivated by the high cost of SVD updates for gradient subspace switching and their observation that some network layers exhibit a high degree of similarity between subsequent gradient subspace updates, the authors propose an *adaptive lazy update* schedule which decouples the subspace switching interval between layers. Further, the authors find that the projection matrices are highly amenable to quantization. Stochastic rounding of the high-rank re-projected gradients is used to reduce information loss when updating the quantized weights.

## Strengths
* This is an important and timely topic as memory efficiency is critical to both the open source / academic community for improving accessibility to LLMs and also to industry where a smaller memory footprint translates directly into lower overhead costs for training and fine-tuning.
* The paper is well motivated based on empirical findings that the authors present related to the similarity between gradient subspaces during the course of a typical pre-training run.
* The paper is well written and easy to follow.
* Figures are clear and enhance the overall work.
* Significant memory savings are achieved in both the pretraining and fine-tuning settings.
* Evaluations demonstrate that Q-GaLore outperforms well considered benchmarks in terms of both memory overhead, PPL for pretraining, and MMLU on fine-tuned models.

## Weaknesses
* Scalability: The largest pretraining model considered was 1B, which is, in my opinion, too small to draw any strong conclusions from w.r.t. foundational model sizes. This is particularly notable as in Table 1 we see a trend where the PPL delta between GaLore and Q-GaLore increases as the model size increases. Likely, this may be due to the int8 weight quantization and associated challenges with training quantized networks; however, it does lead to some concern that Q-GaLore may not perform comparably with GaLore at model sizes of 7-100B.
* Throughput: One of the motivating factors of developing Q-GaLore is acknowledged to be the high computational cost of SVD updates in the original GaLore method. The lazy adaptive update schedule seems to address this in principle by greatly reducing the average number of subspace updates (i.e., Fig 7); however, the authors note that the quantization/dequantization operations actually end up incurring an 14.64% throughput penalty compared to GaLore. Could the authors clarify if this comparison includes the latency required for the more frequent SVD subspace updates in GaLore? If so, it’s a bit disappointing as the additional quantization operations seem to have eliminated the potential direct benefit of the lazy adaptive update schedule. One ablation which may be worth examining by the authors to improve this work would be GaLore+adaptive updates. This may yield a nice result for latency in spite of limited memory benefits. Another approach could be to check if the native FP8 support on Hopper devices would reduce the number of quantization/dequant operations required and thereby close the throughput gap with the original GaLore. However, I agree that this work may be best suited for commodity hardware which would not have such support in any case.
* Limited downstream evaluations: Only MMLU was considered. A more comprehensive downstream evaluation suite such as OpenLLM leaderboard V1 could further establish the benefits/drawbacks of Q-GaLore. Many recent works have noted that abstract reasoning and math performance are specifically negatively impacted by model compression and it would be interesting to note if the same observation applies here.


## Minor concerns
* Typo on L143: occupis -> occupies

---

### Meta-Review · Area_Chair_txbu · 2025-02-04

**Recommendation:** Accept (Poster)
**Confidence:** 4

**Metareview:**

This work presents a quantized memory efficient training method. Most reviewers are voting for acceptance. I would also recommend to compare 8 bit model and 4 bit model, which I guess can be easily implemented using the Bits and Bytes package.

---

### Decision · Program_Chairs · 2025-02-11

Accept (Poster)